# Latent Alignment and Variational Attention

**Yuntian Deng**[*]    **Yoon Kim**[*]    **Justin Chiu**    **Demi Guo**    **Alexander M. Rush**

{dengyuntian@seas,yoonkim@seas,justinchiu@g,dguo@college,srush@seas}.harvard.edu

School of Engineering and Applied Sciences
Harvard University
Cambridge, MA, USA

## Abstract

Neural attention has become central to many state-of-the-art models in natural language processing and related domains. Attention networks are an easy-to-train and effective method for softly simulating alignment; however, the approach does not marginalize over latent alignments in a probabilistic sense. This property makes it difficult to compare attention to other alignment approaches, to compose it with probabilistic models, and to perform posterior inference conditioned on observed data. A related latent approach, hard attention, fixes these issues, but is generally harder to train and less accurate. This work considers *variational attention* networks, alternatives to soft and hard attention for learning latent variable alignment models, with tighter approximation bounds based on amortized variational inference. We further propose methods for reducing the variance of gradients to make these approaches computationally feasible. Experiments show that for machine translation and visual question answering, inefficient exact latent variable models outperform standard neural attention, but these gains go away when using hard attention based training. On the other hand, variational attention retains most of the performance gain but with training speed comparable to neural attention.

## 1  Introduction

Attention networks [6] have quickly become the foundation for state-of-the-art models in natural language understanding, question answering, speech recognition, image captioning, and more [15, 81, 16, 14, 63, 80, 71, 62]. Alongside components such as residual blocks and long-short term memory networks, soft attention provides a rich neural network building block for controlling gradient flow and encoding inductive biases. However, more so than these other components, which are often treated as black-boxes, researchers use intermediate attention decisions directly as a tool for model interpretability [43, 1] or as a factor in final predictions [25, 68]. From this perspective, attention plays the role of a latent alignment variable [10, 37]. An alternative approach, hard attention [80], makes this connection explicit by introducing a latent variable for alignment and then optimizing a bound on the log marginal likelihood using policy gradients. This approach generally performs worse (aside from a few exceptions such as [80]) and is used less frequently than its soft counterpart.

Still the latent alignment approach remains appealing for several reasons: (a) latent variables facilitate reasoning about dependencies in a probabilistically principled way, e.g. allowing composition with other models, (b) posterior inference provides a better basis for model analysis and partial predictions than strictly feed-forward models, which have been shown to underperform on alignment in machine translation [38], and finally (c) directly maximizing marginal likelihood may lead to better results.

---

[*]Equal contribution.

The aim of this work is to quantify the issues with attention and propose alternatives based on recent developments in variational inference. While the connection between variational inference and hard attention has been noted in the literature [4, 41], the space of possible bounds and optimization methods has not been fully explored and is growing quickly. These tools allow us to better quantify whether the general underperformance of hard attention models is due to modeling issues (i.e. soft attention imbues a better inductive bias) or optimization issues.

Our main contribution is a *variational attention* approach that can effectively fit latent alignments while remaining tractable to train. We consider two variants of variational attention: *categorical* and *relaxed*. The categorical method is fit with amortized variational inference using a learned inference network and policy gradient with a soft attention variance reduction baseline. With an appropriate inference network (which conditions on the entire source/target), it can be used at training time as a drop-in replacement for hard attention. The relaxed version assumes that the alignment is sampled from a Dirichlet distribution and hence allows attention over multiple source elements.

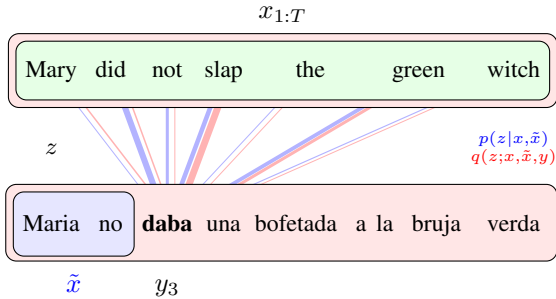

Figure 1: Sketch of variational attention applied to machine translation. Two alignment distributions are shown, the blue prior $p$, and the red variational posterior $q$ taking into account future observations. Our aim is to use $q$ to improve estimates of $p$ and to support improved inference of $z$.

Experiments describe how to implement this approach for two major attention-based models: neural machine translation and visual question answering (Figure 1 gives an overview of our approach for machine translation). We first show that maximizing exact marginal likelihood can increase performance over soft attention. We further show that with variational (categorical) attention, alignment variables significantly surpass both soft and hard attention results without requiring much more difficult training. We further explore the impact of posterior inference on alignment decisions, and how latent variable models might be employed. Our code is available at `https://github.com/harvardnlp/var-attn/`.

**Related Work** Latent alignment has long been a core problem in NLP, starting with the seminal IBM models [11], HMM-based alignment models [75], and a fast log-linear reparameterization of the IBM 2 model [20]. Neural soft attention models were originally introduced as an alternative approach for neural machine translation [6], and have subsequently been successful on a wide range of tasks (see [15] for a review of applications). Recent work has combined neural attention with traditional alignment [18, 72] and induced structure/sparsity [48, 33, 44, 85, 54, 55, 49], which can be combined with the variational approaches outlined in this paper.

In contrast to soft attention models, hard attention [80, 3] approaches use a single sample at training time instead of a distribution. These models have proven much more difficult to train, and existing works typically treat hard attention as a black-box reinforcement learning problem with log-likelihood as the reward [80, 3, 53, 26, 19]. Two notable exceptions are [4, 41]: both utilize amortized variational inference to learn a sampling distribution which is used obtain importance-sampled estimates of the log marginal likelihood [12]. Our method uses uses different estimators and targets the single sample approach for efficiency, allowing the method to be employed for NMT and VQA applications.

There has also been significant work in using variational autoencoders for language and translation application. Of particular interest are those that augment an RNN with latent variables (typically Gaussian) at each time step [17, 22, 66, 23, 40] and those that incorporate latent variables into sequence-to-sequence models [84, 7, 70, 64]. Our work differs by modeling an explicit model component (alignment) as a latent variable instead of auxiliary latent variables (e.g. topics). The term "variational attention" has been used to refer to a different component the output from attention (commonly called the context vector) as a latent variable [7], or to model both the memory and the alignment as a latent variable [9]. Finally, there is some parallel work [78, 67] which also performs exact/approximate marginalization over latent alignments for sequence-to-sequence learning.

## 2 Background: Latent Alignment and Neural Attention

We begin by introducing notation for latent alignment, and then show how it relates to neural attention. For clarity, we are careful to use *alignment* to refer to this probabilistic model (Section 2.1), and *soft* and *hard* attention to refer to two particular inference approaches used in the literature to estimate alignment models (Section 2.2).

### 2.1 Latent Alignment

Figure 2(a) shows a latent alignment model. Let $x$ be an observed set with associated members $\{x_1, \ldots, x_i, \ldots, x_T\}$. Assume these are vector-valued (i.e. $x_i \in \mathbb{R}^d$) and can be stacked to form a matrix $X \in \mathbb{R}^{d \times T}$. Let the observed $\tilde{x}$ be an arbitrary "query". These generate a discrete output variable $y \in \mathcal{Y}$. This process is mediated through a latent alignment variable $z$, which indicates which member (or mixture of members) of $x$ generates $y$. The generative process we consider is:

$$z \sim \mathcal{D}(a(x, \tilde{x}; \theta)) \quad y \sim f(x, z; \theta)$$

where $a$ produces the parameters for an alignment distribution $\mathcal{D}$. The function $f$ gives a distribution over the output, e.g. an exponential family. To fit this model to data, we set the model parameters $\theta$ by maximizing the log marginal likelihood of training examples $(x, \tilde{x}, \hat{y})$:[2]

$$\max_{\theta} \, \log p(y = \hat{y} \,|\, x, \, \tilde{x}) \quad = \quad \max_{\theta} \, \log \mathbb{E}_z[f(x, z; \theta)_{\hat{y}}]$$

Directly maximizing this log marginal likelihood in the presence of the latent variable $z$ is often difficult due to the expectation (though tractable in certain cases).

For this to represent an alignment, we restrict the variable $z$ to be in the simplex $\Delta^{T-1}$ over source indices $\{1, \ldots, T\}$. We consider two distributions for this variable: first, let $\mathcal{D}$ be a *categorical* where $z$ is a one-hot vector with $z_i = 1$ if $x_i$ is selected. For example, $f(x, z)$ could use $z$ to pick from $x$ and apply a softmax layer to predict $y$, i.e. $f(x, z) = \mathrm{softmax}(\mathbf{W}Xz)$ and $\mathbf{W} \in \mathbb{R}^{|\mathcal{Y}| \times d}$,

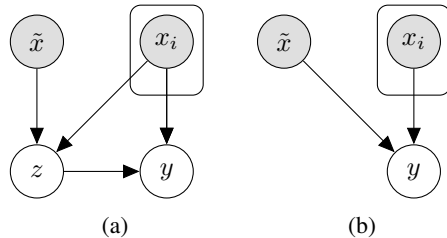

(a)                (b)

Figure 2: Models over observed set $x$, query $\tilde{x}$, and alignment $z$. (a) Latent alignment model, (b) Soft attention with $z$ absorbed into prediction network.

$$\log p(y = \hat{y} \,|\, x, \, \tilde{x}) = \log \sum_{i=1}^{T} p(z_i = 1 \,|\, x, \tilde{x}) p(y = \hat{y} \,|\, x, z_i = 1) = \log \mathbb{E}_z[\mathrm{softmax}(\mathbf{W}Xz)_{\hat{y}}]$$

This computation requires a factor of $O(T)$ additional runtime, and introduces a major computational factor into already expensive deep learning models.[3]

Second we consider a *relaxed* alignment where $z$ is a mixture taken from the interior of the simplex by letting $\mathcal{D}$ be a Dirichlet. This objective looks similar to the categorical case, i.e. $\log p(y = \hat{y} \,|\, x, \, \tilde{x}) = \log \mathbb{E}_z[\mathrm{softmax}(\mathbf{W}Xz)_{\hat{y}}]$, but the resulting expectation is intractable to compute exactly.

### 2.2 Attention Models: Soft and Hard

When training deep learning models with gradient methods, it can be difficult to use latent alignment directly. As such, two alignment-like approaches are popular: *soft attention* replaces the probabilistic model with a deterministic soft function and *hard attention* trains a latent alignment model by maximizing a lower bound on the log marginal likelihood (obtained from Jensen's inequality) with policy gradient-style training. We briefly describe how these methods fit into this notation.

**Soft Attention** Soft attention networks use an altered model shown in Figure 2b. Instead of using a latent variable, they employ a deterministic network to compute an expectation over the alignment variable. We can write this model using the same functions $f$ and $a$ from above,

$$\log p_{\text{soft}}(y \mid x, \ \tilde{x}) = \log f(x, \mathbb{E}_z[z]; \theta) = \log \text{softmax}(\mathbf{W} X \mathbb{E}_z[z])$$

A major benefit of soft attention is efficiency. Instead of paying a multiplicative penalty of $O(T)$ or requiring integration, the soft attention model can compute the expectation before $f$. While formally a different model, soft attention has been described as an approximation of alignment [80]. Since $\mathbb{E}[z] \in \Delta^{T-1}$, soft attention uses a convex combination of the input representations $X\mathbb{E}[z]$ (the *context vector*) to obtain a distribution over the output. While also a "relaxed" decision, this expression differs from both the latent alignment models above. Depending on $f$, the gap between $\mathbb{E}[f(x,z)]$ and $f(x, \mathbb{E}[z])$ may be large.

However there are some important special cases. In the case where $p(z \mid x, \tilde{x})$ is deterministic, we have $\mathbb{E}[f(x,z)] = f(x, \mathbb{E}[z])$, and $p(y \mid x, \ \tilde{x}) = p_{\text{soft}}(y \mid x, \ \tilde{x})$. In general we can bound the absolute difference based on the maximum curvature of $f$, as shown by the following proposition.

**Proposition 1.** *Define* $g_{x,\hat{y}} : \Delta^{T-1} \mapsto [0,1]$ *to be the function given by* $g_{x,\hat{y}}(z) = f(x,z)_{\hat{y}}$ *(i.e.* $g_{x,\hat{y}}(z) = p(y = \hat{y} \mid x, \tilde{x}, z))$ *for a twice differentiable function* $f$. *Let* $H_{g_{x,\hat{y}}}(z)$ *be the Hessian of* $g_{x,\hat{y}}(z)$ *evaluated at* $z$, *and further suppose* $\|H_{g_{x,\hat{y}}}(z)\|_2 \leq c$ *for all* $z \in \Delta^{T-1}, \hat{y} \in \mathcal{Y}$, *and* $x$, *where* $\| \cdot \|_2$ *is the spectral norm. Then for all* $\hat{y} \in \mathcal{Y}$,

$$|\, p(y = \hat{y} \mid x, \tilde{x}) - p_{\text{soft}}(y = \hat{y} \mid x, \tilde{x})\,| \leq c$$

The proof is given in Appendix A.[4] Empirically the soft approximation works remarkably well, and often moves towards a sharper distribution with training. Alignment distributions learned this way often correlate with human intuition (e.g. word alignment in machine translation) [38].[5]

**Hard Attention** Hard attention is an approximate inference approach for latent alignment (Figure 2a) [80, 4, 53, 26]. Hard attention takes a single hard sample of $z$ (as opposed to a soft mixture) and then backpropagates through the model. The approach is derived by two choices: First apply Jensen's inequality to get a lower bound on the log marginal likelihood, $\log \mathbb{E}_z[p(y \mid x, z)] \geq \mathbb{E}_z[\log p(y \mid x, z)]$, then maximize this lower-bound with policy gradients/REINFORCE [76] to obtain unbiased gradient estimates,

$$\nabla_\theta \mathbb{E}_z[\log f(x,z))] = \mathbb{E}_z[\nabla_\theta \log f(x,z) + (\log f(x,z) - B)\nabla_\theta \log p(z \mid x, \tilde{x})],$$

where $B$ is a baseline that can be used to reduce the variance of this estimator. To implement this approach efficiently, hard attention uses Monte Carlo sampling to estimate the expectation in the gradient computation. For efficiency, a single sample from $p(z \mid x, \tilde{x})$ is used, in conjunction with other tricks to reduce the variance of the gradient estimator (discussed more below) [80, 50, 51].

## 3 Variational Attention for Latent Alignment Models

Amortized variational inference (AVI, closely related to variational auto-encoders) [36, 61, 50] is a class of methods to efficiently approximate latent variable inference, using learned inference networks. In this section we explore this technique for deep latent alignment models, and propose methods for *variational attention* that combine the benefits of soft and hard attention.

First note that the key approximation step in hard attention is to optimize a lower bound derived from Jensen's inequality. This gap could be quite large, contributing to poor performance.[6] Variational

| **Algorithm 1** Variational Attention | **Algorithm 2** Variational Relaxed Attention |
|---|---|
| $\lambda \leftarrow \text{enc}(x, \tilde{x}, y; \phi)$ ▷ *Compute var. params* | $\max_\theta \mathbb{E}_{z \sim p}[\log p(y \,|\, x, z)]$ ▷ *Pretrain fixed $\theta$* |
| $z \sim q(z; \lambda)$ ▷ *Sample var. attention* | ... |
| $\log f(x, z)$ ▷ Compute output dist | $u \sim \mathcal{U}$ ▷ *Sample unparam.* |
| $z' \leftarrow \mathbb{E}_{p(z' \,|\, x, \tilde{x})}[z']$ ▷ Compute soft atten. | $z \leftarrow g_\phi(u)$ ▷ *Reparam sample* |
| $B = \log f(x, z')$ ▷ Compute baseline dist | $\log f(x, z)$ ▷ Compute output dist |
| Backprop $\nabla_\theta$ and $\nabla_\phi$ based on eq. 1 and KL | Backprop $\nabla_\theta$ and $\nabla_\phi$, reparam and KL |

inference methods directly aim to tighten this gap. In particular, the *evidence lower bound* (ELBO) is a parameterized bound over a family of distributions $q(z) \in \mathcal{Q}$ (with the constraint that the $\text{supp}\, q(z) \subseteq \text{supp}\, p(z \,|\, x, \tilde{x}, y)$),

$$\log \mathbb{E}_{z \sim p(z \,|\, x, \tilde{x})}[p(y \,|\, x, z)] \geq \mathbb{E}_{z \sim q(z)}[\log p(y \,|\, x, z)] - \text{KL}[q(z) \,\|\, p(z \,|\, x, \tilde{x})]$$

This allows us to search over variational distributions $q$ to improve the bound. It is tight when the variational distribution is equal to the posterior, i.e. $q(z) = p(z \,|\, x, \tilde{x}, y)$. Hard attention is a special case of the ELBO with $q(z) = p(z \,|\, x, \tilde{x})$.

There are many ways to optimize the evidence lower bound; an effective choice for deep learning applications is to use *amortized variational inference*. AVI uses an *inference network* to produce the parameters of the variational distribution $q(z; \lambda)$. The inference network takes in the input, query, and the output, i.e. $\lambda = enc(x, \tilde{x}, y; \phi)$. The objective aims to reduce the gap with the inference network $\phi$ while also training the generative model $\theta$,

$$\max_{\phi, \theta} \mathbb{E}_{z \sim q(z; \lambda)}[\log p(y \,|\, x, z)] - \text{KL}[q(z; \lambda) \,\|\, p(z \,|\, x, \tilde{x})]$$

With the right choice of optimization strategy and inference network this form of variational attention can provide a general method for learning latent alignment models. In the rest of this section, we consider strategies for accurately and efficiently computing this objective; in the next section, we describe instantiations of $enc$ for specific domains.

**Algorithm 1: Categorical Alignments** First consider the case where $\mathcal{D}$, the alignment distribution, and $\mathcal{Q}$, the variational family, are categorical distributions. Here the generative assumption is that $y$ is generated from a single index of $x$. Under this setup, a low-variance estimator of $\nabla_\theta \text{ELBO}$, is easily obtained through a single sample from $q(z)$. For $\nabla_\phi \text{ELBO}$, the gradient with respect to the KL portion is easily computable, but there is an optimization issue with the gradient with respect to the first term $\mathbb{E}_{z \sim q(z)}[\log f(x, z))]$.

Many recent methods target this issue, including neural estimates of baselines [50, 51], Rao-Blackwellization [59], reparameterizable relaxations [31, 47], and a mix of various techniques [73, 24]. We found that an approach using REINFORCE [76] along with a specialized baseline was effective. However, note that REINFORCE is only one of the inference choices we can select, and as we will show later, alternative approaches such as reparameterizable relaxations work as well. Formally, we first apply the likelihood-ratio trick to obtain an expression for the gradient with respect to the inference network parameters $\phi$,

$$\nabla_\phi \mathbb{E}_{z \sim q(z)}[\log p(y \,|\, x, z)] = \mathbb{E}_{z \sim q(z)}[(\log f(x, z) - B)\nabla_\phi \log q(z)]$$

As with hard attention, we take a single Monte Carlo sample (now drawn from the variational distribution). Variance reduction of this estimate falls to the baseline term $B$. The ideal (and intuitive) baseline would be $\mathbb{E}_{z \sim q(z)}[\log f(x, z)]$, analogous to the value function in reinforcement learning. While this term cannot be easily computed, there is a natural, cheap approximation: soft attention (i.e. $\log f(x, \mathbb{E}[z])$). Then the gradient is

$$\mathbb{E}_{z \sim q(z)} \left[ \left( \log \frac{f(x, z)}{f(x, \mathbb{E}_{z' \sim p(z' \,|\, x, \tilde{x})}[z'])} \right) \nabla_\phi \log q(z \,|\, x, \tilde{x}) \right] \tag{1}$$

Effectively this weights gradients to $q$ based on the ratio of the inference network alignment approach to a soft attention baseline. Notably the expectation in the soft attention is over $p$ (and not over $q$), and therefore the baseline is constant with respect to $\phi$. Note that a similar baseline can also be used for hard attention, and we apply it to both variational/hard attention models in our experiments.

**Algorithm 2: Relaxed Alignments**  Next consider treating both $\mathcal{D}$ and $\mathcal{Q}$ as Dirichlets, where $z$ represents a mixture of indices. This model is in some sense closer to the soft attention formulation which assigns mass to multiple indices, though fundamentally different in that we still formally treat alignment as a latent variable. Again the aim is to find a low variance gradient estimator. Instead of using REINFORCE, certain continuous distributions allow the use reparameterization [36], where sampling $z \sim q(z)$ can be done by first sampling from a simple unparameterized distribution $\mathcal{U}$, and then applying a transformation $g_\phi(\cdot)$, yielding an unbiased estimator,

$$\mathbb{E}_{u \sim \mathcal{U}}\left[\nabla_\phi \log p(y|x, g_\phi(u))\right] - \nabla_\phi \operatorname{KL}\left[q(z) \,\|\, p(z \,|\, x, \tilde{x})\right]$$

The Dirichlet distribution is not directly reparameterizable. While transforming the standard uniform distribution with the inverse CDF of Dirichlet would result in a Dirichlet distribution, the inverse CDF does not have an analytical solution. However, we can use rejection based sampling to get a sample, and employ implicit differentiation to estimate the gradient of the CDF [32].

Empirically, we found the random initialization would result in convergence to uniform Dirichlet parameters for $\lambda$. (We suspect that it is easier to find low KL local optima towards the center of the simplex). In experiments, we therefore initialize the latent alignment model by first minimizing the Jensen bound, $\mathbb{E}_{z \sim p(z \,|\, x, \tilde{x})}[\log p(y \,|\, x, z)]$, and then introducing the inference network.

## 4   Models and Methods

We experiment with variational attention in two different domains where attention-based models are essential and widely-used: neural machine translation and visual question answering.

**Neural Machine Translation**  Neural machine translation (NMT) takes in a source sentence and predicts each word of a target sentence $y_j$ in an auto-regressive manner. The model first contextually embeds each source word using a bidirectional LSTM to produce the vectors $x_1 \ldots x_T$. The query $\tilde{x}$ consists of an LSTM-based representation of the previous target words $y_{1:j-1}$. Attention is used to identify which source positions should be used to predict the target. The parameters of $\mathcal{D}$ are generated from an MLP between the query and source [6], and $f$ concatenates the selected $x_i$ with the query $\tilde{x}$ and passes it to an MLP to produce the distribution over the next target word $y_j$.

For variational attention, the inference network applies a bidirectional LSTM over the source and the target to obtain the hidden states $x_1, \ldots, x_T$ and $h_1, \ldots, h_S$, and produces the alignment scores at the $j$-th time step via a bilinear map, $s_i^{(j)} = \exp(h_j^\top \mathbf{U} x_i)$. For the categorical case, the scores are normalized, $q(z_i^{(j)} = 1) \propto s_i^{(j)}$; in the relaxed case the parameters of the Dirichlet are $\alpha_i^{(j)} = s_i^{(j)}$. Note, the inference network sees the entire target (through bidirectional LSTMs). The word embeddings are shared between the generative/inference networks, but other parameters are separate.

**Visual Question Answering**  Visual question answering (VQA) uses attention to locate the parts of an image that are necessary to answer a textual question. We follow the recently-proposed "bottom-up top-down" attention approach [2], which uses Faster R-CNN [60] to obtain object bounding boxes and performs mean-pooling over the convolutional features (from a pretrained ResNet-101 [27]) in each bounding box to obtain object representations $x_1, \ldots, x_T$. The query $\tilde{x}$ is obtained by running an LSTM over the question, the attention function $a$ passes the query and the object representation through an MLP. The prediction function $f$ is also similar to the NMT case: we concatenate the chosen $x_i$ with the query $\tilde{x}$ to use as input to an MLP which produces a distribution over the output. The inference network $enc$ uses the answer embedding $h_y$ and combines it with $x_i$ and $\tilde{x}$ to produce the variational (categorical) distribution,

$$q(z_i = 1) \propto \exp(u^\top \tanh(\mathbf{U}_1(x_i \odot \operatorname{ReLU}(\mathbf{V}_1 h_y)) + \mathbf{U}_2(\tilde{x} \odot \operatorname{ReLU}(\mathbf{V}_2 h_y))))$$

where $\odot$ is the element-wise product. This parameterization worked better than alternatives. We did not experiment with the relaxed case in VQA, as the object bounding boxes already give us the ability to attend to larger portions of the image.

**Inference Alternatives**  For categorical alignments we described maximizing a particular variational lower bound with REINFORCE. Note that other alternatives exist, and we briefly discuss them

here: 1) instead of the single-sample variational bound we can use a multiple-sample importance sampling based approach such as Reweighted Wake-Sleep (RWS) [4] or VIMCO [52]; 2) instead of REINFORCE we can approximate sampling from the discrete categorical distribution with Gumbel-Softmax [30]; 3) instead of using an inference network we can directly apply Stochastic Variational Inference (SVI) [28] to learn the local variational parameters in the posterior.

**Predictive Inference**   At test time, we need to marginalize out the latent variables, i.e. $\mathbb{E}_z[p(y \mid x, \tilde{x}, z)]$ using $p(z \mid x, \tilde{x})$. In the categorical case, if speed is not an issue then enumerating alignments is preferable, which incurs a multiplicative cost of $O(T)$ (but the enumeration is parallelizable). Alternatively we experimented with a $K$-max renormalization, where we only take the top-$K$ attention scores to approximate the attention distribution (by re-normalizing). This makes the multiplicative cost constant with respect to $T$. For the relaxed case, sampling is necessary.

# 5   Experiments

**Setup**   For NMT we mainly use the IWSLT dataset [13]. This dataset is relatively small, but has become a standard benchmark for experimental NMT models. We follow the same preprocessing as in [21] with the same Byte Pair Encoding vocabulary of 14k tokens [65]. To show that variational attention scales to large datasets, we also experiment on the WMT 2017 English-German dataset [8], following the preprocessing in [74] except that we use newstest2017 as our test set. For VQA, we use the VQA 2.0 dataset. As we are interested in intrinsic evaluation (i.e. log-likelihood) in addition to the standard VQA metric, we randomly select half of the standard validation set as the test set (since we need access to the actual labels).[7] (Therefore the numbers provided are not strictly comparable to existing work.) While the preprocessing is the same as [2], our numbers are worse than previously reported as we do not apply any of the commonly-utilized techniques to improve performance on VQA such as data augmentation and label smoothing.

Experiments vary three components of the systems: (a) training objective and model, (b) training approximations, comparing enumeration or sampling,[8] (c) test inference. All neural models have the same architecture and the exact same number of parameters $\theta$ (the inference network parameters $\phi$ vary, but are not used at test). When training hard and variational attention with sampling both use the same baseline, i.e the output from soft attention. The full architectures/hyperparameters for both NMT and VQA are given in Appendix B.

**Results and Discussion**   Table 1 shows the main results. We first note that hard attention underperforms soft attention, even when its expectation is enumerated. This indicates that Jensen's inequality alone is a poor bound. On the other hand, on both experiments, exact marginal likelihood outperforms soft attention, indicating that when possible it is better to have latent alignments.

For NMT, on the IWSLT 2014 German-English task, variational attention with enumeration and sampling performs comparably to optimizing the log marginal likelihood, despite the fact that it is optimizing a lower bound. We believe that this is due to the use of $q(z)$, which conditions on the entire source/target and therefore potentially provides better training signal to $p(z \mid x, \tilde{x})$ through the KL term. Note that it is also possible to have $q(z)$ come from a pretrained external model, such as a traditional alignment model [20]. Table 3 (left) shows these results in context compared to the best reported values for this task. Even with sampling, our system improves on the state-of-the-art. On the larger WMT 2017 English-German task, the superior performance of variational attention persists: our baseline soft attention reaches 24.10 BLEU score, while variational attention reaches 24.98. Note that this only reflects a reasonable setting without exhaustive tuning, yet we show that we can train variational attention at scale. For VQA the trend is largely similar, and results for NLL with variational attention improve on soft attention and hard attention. However the task-specific evaluation metrics are slightly worse.

Table 2 (left) considers test inference for variational attention, comparing enumeration to $K$-max with $K = 5$. For all methods exact enumeration is better, however $K$-max is a reasonable approximation.

| Model | Objective | $\mathbb{E}$ | NMT PPL | NMT BLEU | VQA NLL | VQA Eval |
|---|---|---|---|---|---|---|
| Soft Attention | $\log p(y \mid \mathbb{E}[z])$ | - | 7.17 | 32.77 | 1.76 | 58.93 |
| Marginal Likelihood | $\log \mathbb{E}[p]$ | Enum | 6.34 | 33.29 | 1.69 | 60.33 |
| Hard Attention | $\mathbb{E}_p[\log p]$ | Enum | 7.37 | 31.40 | 1.78 | 57.60 |
| Hard Attention | $\mathbb{E}_p[\log p]$ | Sample | 7.38 | 31.00 | 1.82 | 56.30 |
| Variational Relaxed Attention | $\mathbb{E}_q[\log p] - \mathrm{KL}$ | Sample | 7.58 | 30.05 | - | - |
| Variational Attention | $\mathbb{E}_q[\log p] - \mathrm{KL}$ | Enum | 6.08 | 33.68 | 1.69 | 58.44 |
| Variational Attention | $\mathbb{E}_q[\log p] - \mathrm{KL}$ | Sample | 6.17 | 33.30 | 1.75 | 57.52 |

Table 1: Evaluation on NMT and VQA for the various models. $\mathbb{E}$ column indicates whether the expectation is calculated via enumeration (Enum) or a single sample (Sample) during training. For NMT we evaluate intrinsically on perplexity (PPL) (lower is better) and extrinsically on BLEU (higher is better), where for BLEU we perform beam search with beam size 10 and length penalty (see Appendix B for further details). For VQA we evaluate intrinsically on negative log-likelihood (NLL) (lower is better) and extrinsically on VQA evaluation metric (higher is better). All results except for relaxed attention use enumeration at test time.

| Model | PPL Exact | PPL $K$-Max | BLEU Exact | BLEU $K$-Max |
|---|---|---|---|---|
| Marginal Likelihood | 6.34 | 6.90 | 33.29 | 33.31 |
| Hard + Enum | 7.37 | 7.37 | 31.40 | 31.37 |
| Hard + Sample | 7.38 | 7.38 | 31.00 | 31.04 |
| Variational + Enum | 6.08 | 6.42 | 33.68 | 33.69 |
| Variational + Sample | 6.17 | 6.51 | 33.30 | 33.27 |

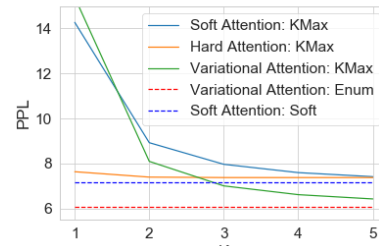

Table 2: (Left) Performance change on NMT from exact decoding to $K$-Max decoding with $K = 5$. (see section 5 for definition of K-max decoding). (Right) Test perplexity of different approaches while varying $K$ to estimate $\mathbb{E}_z[p(y|x,\tilde{x})]$. Dotted lines compare soft baseline and variational with full enumeration.

Table 2 (right) shows the PPL of different models as we increase $K$. Good performance requires $K > 1$, but we only get marginal benefits for $K > 5$. Finally, we observe that it is possible to *train* with soft attention and *test* using $K$-Max with a small performance drop (`Soft KMax` in Table 2 (right)). This possibly indicates that soft attention models are approximating latent alignment models. On the other hand, training with latent alignments and testing with soft attention performed badly.

Table 3 (lower right) looks at the entropy of the prior distribution learned by the different models. Note that hard attention has very low entropy (high certainty) whereas soft attention is quite high. The variational attention model falls in between. Figure 3 (left) illustrates the difference in practice.

Table 3 (upper right) compares inference alternatives for variational attention. RWS reaches a comparable performance as REINFORCE, but at a higher memory cost as it requires multiple samples. Gumbel-Softmax reaches nearly the same performance and seems like a viable alternative; although we found its performance is sensitive to its temperature parameter. We also trained a non-amortized SVI model, but found that at similar runtime it was not able to produce satisfactory results, likely due to insufficient updates of the local variational parameters. A hybrid method such as semi-amortized inference [39, 34] might be a potential future direction worth exploring.

Despite extensive experiments, we found that variational relaxed attention performed worse than other methods. In particular we found that when training with a Dirichlet KL, it is hard to reach low-entropy regions of the simplex, and the attentions are more uniform than either soft or variational categorical attention. Table 3 (lower right) quantifies this issue. We experimented with other distributions such as Logistic-Normal and Gumbel-Softmax [31, 47] but neither fixed this issue. Others have also noted difficulty in training Dirichlet models with amortized inference [69].

Besides performance, an advantage of these models is the ability to perform posterior inference, since the $q$ function can be used directly to obtain posterior alignments. Contrast this with hard attention where $q = p(z \mid x, \tilde{x})$, i.e. the variational posterior is independent of the future information. Figure 3 shows the alignments of $p$ and $q$ for variational attention over a fixed sentence (see Appendix C for more examples). We see that $q$ is able to use future information to correct alignments. We note that the inability of soft and hard attention to produce good alignments has been noted as a major issue in NMT [38]. While $q$ is not used directly in left-to-right NMT decoding, it could be employed for other applications such as in an iterative refinement approach [56, 42].

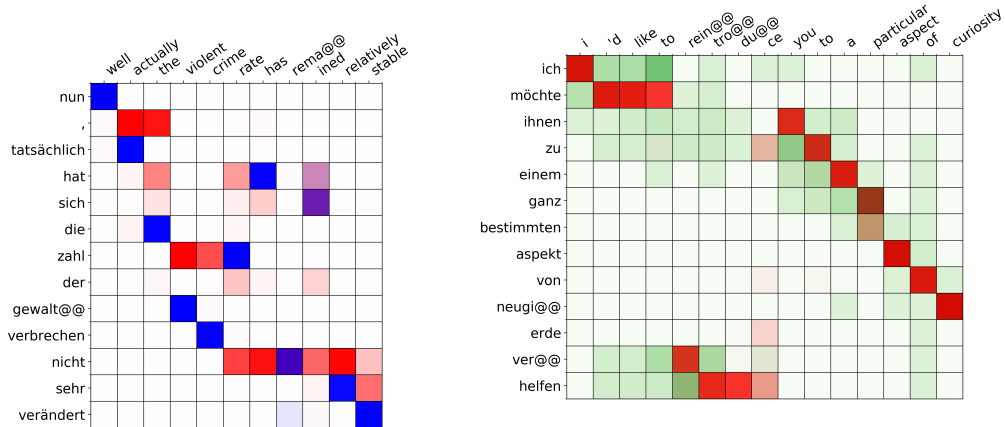

Figure 3: (Left) An example demonstrating the difference between the prior alignment (red) and the variational posterior (blue) when translating from DE-EN (left-to-right). Note the improved blue alignments for `actually` and `violent` which benefit from seeing the next word. (Right) Comparison of soft attention (green) with the $p$ of variational attention (red). Both models imply a similar alignment, but variational attention has lower entropy.

| Model | IWSLT BLEU |
|---|---|
| Beam Search Optimization [77] | 26.36 |
| Actor-Critic [5] | 28.53 |
| Neural PBMT + LM [29] | 30.08 |
| Minimum Risk Training [21] | 32.84 |
| Soft Attention | 32.77 |
| Marginal Likelihood | 33.29 |
| Hard Attention + Enum | 31.40 |
| Hard Attention + Sample | 30.42 |
| Variational Relaxed Attention | 30.05 |
| Variational Attention + Enum | 33.69 |
| Variational Attention + Sample | 33.30 |

| Inference Method | #Samples | PPL | BLEU |
|---|---|---|---|
| REINFORCE | 1 | 6.17 | 33.30 |
| RWS | 5 | 6.41 | 32.96 |
| Gumbel-Softmax | 1 | 6.51 | 33.08 |

| Model | Entropy | |
|---|---|---|
| | NMT | VQA |
| Soft Attention | 1.24 | 2.70 |
| Marginal Likelihood | 0.82 | 2.66 |
| Hard Attention + Enum | 0.05 | 0.73 |
| Hard Attention + Sample | 0.07 | 0.58 |
| Variational Relaxed Attention | 2.02 | - |
| Variational Attention + Enum | 0.54 | 2.07 |
| Variational Attention + Sample | 0.52 | 2.44 |

Table 3: (Left) Comparison against the best prior work for NMT on the IWSLT 2014 German-English test set. (Upper Right) Comparison of inference alternatives of variational attention on IWSLT 2014. (Lower Right) Comparison of different models in terms of implied discrete entropy (lower = more certain alignment).

**Potential Limitations** While this technique is a promising alternative to soft attention, there are some practical limitations: (a) Variational/hard attention needs a good baseline estimator in the form of soft attention. We found this to be a necessary component for adequately training the system. This may prevent this technique from working when $T$ is intractably large and soft attention is not an option. (b) For some applications, the model relies heavily on having a good posterior estimator. In VQA we had to utilize domain structure for the inference network construction. (c) Recent models such as the Transformer [74], utilize many repeated attention models. For instance the current best translation models have the equivalent of 150 different attention queries per word translated. It is unclear if this approach can be used at that scale as predictive inference becomes combinatorial.

# 6 Conclusion

Attention methods are ubiquitous tool for areas like natural language processing; however they are difficult to use as latent variable models. This work explores alternative approaches to latent alignment, through variational attention with promising result. Future work will experiment with scaling the method on larger-scale tasks and in more complex models, such as multi-hop attention models, transformer models, and structured models, as well as utilizing these latent variables for interpretability and as a way to incorporate prior knowledge.

## Acknowledgements

We are grateful to Sam Wiseman and Rachit Singh for insightful comments and discussion, as well as Christian Puhrsch for help with translations. This project was supported by a Facebook Research Award (Low Resource NMT). YK is supported by a Google AI PhD Fellowship. YD is supported by a Bloomberg Research Award. AMR gratefully acknowledges the support of NSF CCF-1704834 and an Amazon AWS Research award.

## Footnotes

[2]When clear from context, the random variable is dropped from $\mathbb{E}[\cdot]$. We also interchangeably use $p(\hat{y} \,|\, x, \tilde{x})$ and $f(x, z; \theta)_{\hat{y}}$ to denote $p(y = \hat{y} \,|\, x, \tilde{x})$.

[3]Although not our main focus, explicit marginalization is sometimes tractable with efficient matrix operations on modern hardware, and we compare the variational approach to explicit enumeration in the experiments. In some cases it is also possible to efficiently perform exact marginalization with dynamic programming if one imposes additional constraints (e.g. monotonicity) on the alignment distribution [83, 82, 58].

[4]It is also possible to study the gap in finer detail by considering distributions over the inputs of $f$ that have high probability under approximately linear regions of $f$, leading to the notion of *approximately expectation-linear* functions, which was originally proposed and studied in the context of dropout [46].

[5]Another way of viewing soft attention is as simply a non-probabilistic learned function. While it is possible that such models encode better inductive biases, our experiments show that when properly optimized, latent alignment attention with explicit latent variables do outperform soft attention.

[6]Prior works on hard attention have generally approached the problem as a black-box reinforcement learning problem where the rewards are given by $\log f(x,z)$. Ba et al. (2015) [4] and Lawson et al. (2017) [41] are the notable exceptions, and both works utilize the framework from [51] which obtains multiple samples from a learned sampling distribution to optimize the IWAE bound [12] or a reweighted wake-sleep objective.

[7] VQA eval metric is defined as $\min\{\frac{\text{\# humans that said answer}}{3}, 1\}$. Also note that since there are sometimes multiple answers for a given question, in such cases we sample (where the sampling probability is proportional to the number of humans that said the answer) to get a single label.

[8] Note that enumeration does not imply exact if we are enumerating an expectation on a lower bound.

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
