[Supplementary Material]

# Supplementary Materials for
# Latent Alignment and Variational Attention

## Appendix A: Proof of Proposition 1

**Proposition.** *Define $g_{x,\hat{y}} : \Delta^{T-1} \mapsto [0,1]$ to be the function given by $g_{x,\hat{y}}(z) = f(x,z)_{\hat{y}}$ (i.e. $g_{x,\hat{y}}(z) = p(y = \hat{y} \mid x, \tilde{x}, z))$ for a twice differentiable function $f$. Let $H_{g_{x,\hat{y}}}(z)$ be the Hessian of $g_{x,\hat{y}}(z)$ evaluated at $z$, and further suppose $\|H_{g_{x,\hat{y}}}(z)\|_2 \leq c$ for all $z \in \Delta^{T-1}, \hat{y} \in \mathcal{Y}$, and $x$, where $\|\cdot\|_2$ is the spectral norm. Then for all $\hat{y} \in \mathcal{Y}$,*

$$|\, p(y = \hat{y} \mid x, \tilde{x}) - p_{\text{soft}}(y = \hat{y} \mid x, \tilde{x})\,| \leq c$$

*Proof.* We begin by performing Taylor's expansion of $g_{x,\hat{y}}$ at $\mathbb{E}[z]$:

$$\mathbb{E}[g_{x,\hat{y}}(z)] = \mathbb{E}\Big[g_{x,\hat{y}}(\mathbb{E}[z]) + (z - \mathbb{E}[z])^\top \nabla g_{x,\hat{y}}(\mathbb{E}[z]) + \frac{1}{2}(z - \mathbb{E}[z])^\top H_{g_{x,\hat{y}}}(\hat{z})(z - \mathbb{E}[z])\Big]$$

$$= g_{x,\hat{y}}(\mathbb{E}[z]) + \frac{1}{2}\mathbb{E}[(z - \mathbb{E}[z])^\top H_{g_{x,\hat{y}}}(\hat{z})(z - \mathbb{E}[z])]$$

for some $\hat{z} = \lambda z + (1 - \lambda)\mathbb{E}[z], \lambda \in [0,1]$. Then letting $u = z - \mathbb{E}[z]$, we have

$$|\,(z - \mathbb{E}[z])^\top H_{g_{x,\hat{y}}}(\hat{z})(z - \mathbb{E}[z])\,| = |\, \|u\|_2^2 \frac{u^\top}{\|u\|_2} H_{g_{x,\hat{y}}}(\hat{z}) \frac{u}{\|u\|_2}\,|$$

$$\leq \|u\|_2^2\, c$$

where $c = \max\{|\lambda_{\max}|, |\lambda_{\min}|\}$ is the largest absolute eigenvalue of $H_{g_{x,\hat{y}}}(\hat{z})$. (Here $\lambda_{\max}$ and $\lambda_{\min}$ are maximum/minimum eigenvalues of $H_{g_{X,q}}(\hat{z})$). Note that $c$ is also equal to the spectral norm $\|H_{g_{X,q}}(\hat{z})\|_2$ since the Hessian is symmetric.

Then,

$$|\, \mathbb{E}[(z - \mathbb{E}[z])^\top H_{g_{x,\hat{y}}}(\hat{z})(z - \mathbb{E}[z])]\,| \leq \mathbb{E}[\,|(z - \mathbb{E}[z])^\top H_{g_{x,\hat{y}}}(\hat{z})(z - \mathbb{E}[z])\,|\,]$$

$$\leq \mathbb{E}[\|u\|_2^2 c]$$

$$\leq 2c$$

Here the first inequality follows due to the convexity of the absolute value function and the last inequality follows since

$$\|u\|_2^2 = (z - \mathbb{E}[z])^\top (z - \mathbb{E}[z])$$

$$= z^\top z + \mathbb{E}[z]^\top \mathbb{E}[z] - 2\mathbb{E}[z]^\top z$$

$$\leq z^\top z + \mathbb{E}[z]^\top \mathbb{E}[z]$$

$$\leq 2$$

where the last two inequalities are due to the fact that $z, \mathbb{E}[z] \in \Delta^{T-1}$. Then putting it all together we have,

$$|\, p(y = \hat{y} \mid x, \tilde{x}) - p_{\text{soft}}(y = \hat{y} \mid x, \tilde{x})\,| = |\, \mathbb{E}[g_{x,\hat{y}}(z)] - g_{x,\hat{y}}(\mathbb{E}[z])\,|$$

$$= \frac{1}{2}|\, \mathbb{E}[(z - \mathbb{E}[z])^\top H_{g_{x,\hat{y}}}(\hat{z})(z - \mathbb{E}[z])]\,|$$

$$\leq c$$

$\square$

## Appendix B: Experimental Setup

**Neural Machine Translation**

For data processing we closely follow the setup in [21], which uses Byte Pair Encoding over the combined source/target training set to obtain a vocabulary size of 14,000 tokens. However, different from [21] which uses maximum sequence length of 175, for faster training we only train on sequences of length up to 125.

The encoder is a two-layer bi-directional LSTM with 512 units in each direction, and the decoder as a two-layer LSTM with with 768 units. For the decoder, the convex combination of source hidden states at each time step from the attention distribution is used as additional input at the next time step. Word embedding is 512-dimensional.

The inference network consists of two bi-directional LSTMs (also two-layer and 512-dimensional each) which is run over the source/target to obtain the hidden states at each time step. These hidden states are combined using bilinear attention [45] to produce the variational parameters. (In contrast the generative model uses MLP attention from [6], though we saw little difference between the two parameterizations). Only the word embedding is shared between the inference network and the generative model.

Other training details include: batch size of 6, dropout rate of 0.3, parameter initialization over a uniform distribution $\mathcal{U}[-0.1, 0.1]$, gradient norm clipping at 5, and training for 30 epochs with Adam (learning rate = 0.0003, $\beta_1 = 0.9$, $\beta_2 = 0.999$) [35] with a learning rate decay schedule which starts halving the learning rate if validation perplexity does not improve. Most models converged well before 30 epochs.

For decoding we use beam search with beam size 10 and length penalty $\alpha = 1$, from [79]. The length penalty added about 0.5 BLEU points across all the models.

**Visual Question Answering**

The model first obtains object features by mean-pooling the pretrained ResNet-101 features [27] (which are 2048-dimensional) over object regions given by Faster R-CNN [60].The ResNet features are kept fixed and not fine-tuned during training. We fix the maximum number of possible regions to be 36. For the question embedding we use a one-layer LSTM with 1024 units over word embeddings. The word embeddings are 300-dimensional and initialized with GloVe [57]. The generative model produces a distribution over the possible objects via applying MLP attention, i.e.

$$p(z_i = 1 \,|\, x, \tilde{x}) \propto \exp(w^\top \tanh(\mathbf{W}_1 x_i + \mathbf{W}_2 \tilde{x}))$$

The selected image region is concatenated with the question embedding and fed to a one-layer MLP with ReLU non-linearity and 1024 hidden units.

The inference network produces a categorical distribution over the image regions by interacting the answer embedding $h_y$ (which are 256-dimensional and initialized randomly) with the question embedding $\tilde{x}$ and the image regions $x_i$,

$$q(z_i = 1) \propto \exp(u^\top \tanh(\mathbf{U}_1(x_i \odot \mathrm{ReLU}(\mathbf{V}_1 h_y)) + \mathbf{U}_2(\tilde{x} \odot \mathrm{ReLU}(\mathbf{V}_2 h_y))))$$

where $\odot$ denotes element-wise multiplication. The generative/inference attention MLPs have 1024 hidden units each (i.e. $w, u \in \mathbb{R}^{1024}$).

Other training details include: batch size of 512, dropout rate of 0.5 on the penultimate layer (i.e. before affine transformation into answer vocabulary), and training for 50 epochs with with Adam (learning rate = 0.0005, $\beta_1 = 0.9$, $\beta_2 = 0.999$) [35].

In cases where there is more than one answer for a given question/image pair, we randomly sample the answer, where the sampling probability is proportional to the number of humans who gave the answer.

# Appendix C: Additional Visualizations

(a)

(b)

(c)

(d)

(e)

(f)

Figure 4: (Left Column) Further examples highlighting the difference between the prior alignment (red) and the variational posterior (blue) when translating from DE-EN (left-to-right). The variational posterior is able to better handle reordering; in (a) the variational posterior successfully aligns 'turning' to 'verwandelt', in (c) we see a similar pattern with the alignment of the clause 'that's my brand' to 'das ist meine marke'. In (e) the prior and posterior both are confused by the '-ial' in 'territor-ial', however the posterior still remains more accurate overall and correctly aligns the rest of 'revierverhalten' to 'territorial behaviour'. (Right Column) Additional comparisons between soft attention (green) and the prior alignments of variational attention (red). Alignments from both models are similar, but variational attention is lower entropy. Both soft and variational attention rely on aligning the inserted English word 'orientation' to the comma in (b) since a direct translation does not appear in the German source.