[Reviews · NeurIPS 2018]

Reviewer 1



Update based on author rebuttal: I believe the authors have addressed the main criticisms of this paper (not clear how it's different from prior work) and have also provided additional experiments. I've raised my score accordingly. This paper focuses on using variational inference to train models with a "latent" (stochastic) attention mechanism. They consider attention as a categorical or dirichlet random variable, and explore using posterior inference on alignment decisions. They test various approaches on NMT and visual question answering datasets. Overall, this was an interesting paper exploring some unusual methods for training models with stochastic/discrete attention. As discussed by the paper, the main reasons to explore this approach are 1) the possibility of improved performance and 2) better interpretability. The paper achieves 1) to a reasonable degree - in both NMT and VQA the latent attention-based models are able to improve over standard softmax attention, though to a limited degree. It's not possible to compare the NMT results to most results in the literature as the experiments were done on IWSLT instead of WMT, but this paper is more structured as an exploration of different ways to train latent attention models rather than an attempt at state-of-the-art. I don't think the paper suitably addresses 2). Only Figure 2 really explores whether the alignments are more interpretable, and only to a limited degree. In prior work (Xu et al.) it was shown that while hard attention can produce better results, the alignments can actually be *less* interpretable. I would urge the authors to either more rigorously back up their claim that hard attention is more interpretable, or temper their claims somewhat. Finally, in various places in the paper it is pointed out that Ba et al. and Lawson et al. have both previously explored variational inference approaches for training attention-based models, but no direct comparison to those techniques is presented. If any of the methods explored subsume those approaches and I just missed this connection, then I would suggest making the connection stronger and more broadly providing a better comparison of the approaches and results in this paper to those in prior work. Some minor comments: - "While more principled..." Is it really more principled (as if there's a definition for "more principled")? I find this statement somewhat objectionable. There are many neural network mechanisms which improve performance or are genuinely useful for reasons separate from whether they can be treated as a latent variable. I think the primary reason people use attention is because it's helpful (probably because of gradient dynamics or otherwise), and it just so happens to be interpretable too. People also use model weights, activations, etc. for interpretability reasons but this doesn't imply it would be more principled (or helpful) to treat them as latent variables. - "may improve system" -> "may improve systems" - "Our method uses uses different" -> "Our method uses different" - "f(x, z) could use z to pick from x" Wouldn't it make more sense to say "f(x, z) could use z to pick from X"? - "this may does prevent" -> "may prevent" - "domain structure to for the" -> "domain structure for the"

Reviewer 2



This work introduces variational alternatives to soft and hard attention models, ubiquitous components in modern deep learning architectures. The authors clearly motivate why attention can be considered an approximation of a latent alignment graphical model, show why exact inference of this model is difficult, and propose two variational methods for training such a latent alignment model. Their experiments show that a latent-alignment treatment of attention, trained using marginal log-likelihood or their variational methods, outperforms state-of-the-art baselines for neural MT and VQA. For a researcher who's accustomed to using soft attention without thinking about an underlying graphical model, the paper was a very helpful new perspective; the argument from the point that "for general functions f, the gap between E[f(x, z)] and f(x, E[z]) may be large" was particularly compelling. One thing that I was looking for but didn't find was a quantitative claim about the efficiency benefits of the variational approach as compared to exact marginal likelihood training. It's a factor of O(N) in principle, but to what extent is that hidden by available parallelism in the implementation? I would also ideally like to see the method evaluated in the context of contemporary self-attention architectures like the Transformer, especially because it wasn't immediately clear to me how the approach works when the "enc" function includes additional layers of attention.

Reviewer 3



This paper uses variational approach to learn latent alignment, and evaluated its result on machine translation and visual question answering. The evaluation results show that their approach performs better than conventional attention mechanism. However, the general idea has been explored in [18, 19], and the proposed approach is very similar to existing work, which limits its novelty. The authors briefly described [18, 19], but there is no theoretical or empirical comparisons which makes it unclear whether the proposed method improves existing approaches.